# Levitation and dynamics of bodies in supersaturated fluids

Saverio E. Spagnolie [1,2] ✉, Samuel Christianson[1] & Carsen Grote[1]

A body immersed in a supersaturated fluid like carbonated water can accumulate a dynamic field of bubbles upon its surface. If the body is mobile, the attached bubbles can lift it upward against gravity, but a fluid-air interface can clean the surface of these lifting agents and the body may plummet. The process then begins anew, and continues for as long as the concentration of gas in the fluid supports it. In this work, experiments using fixed and free immersed bodies reveal fundamental features of force development and gas escape. A continuum model which incorporates the dynamics of a surface buoyancy field is used to predict the ranges of body mass and size, and fluid properties, for which the system is most dynamic, and those for which body excursions are suppressed. Simulations are then used to probe systems which are dominated by a small number of large bubbles. Body rotations at the surface are critical for driving periodic vertical motions of large bodies, which in turn can produce body wobbling, rolling, and damped surface 'bouncing' dynamics.

A fluid containing dissolved gas may become supersaturated upon a rapid change in temperature or pressure, leading to bubble formation and eventual gas escape. This phenomenon is most commonly observed when opening a can of sparkling water, or other carbonated beverages. When the fluid is under sufficient pressure, gas accumulation is inhibited, preventing bubble formation[1]. Upon a rapid reduction of pressure, bubbles form on or near any containing or immersed surfaces, then detach and depart towards the fluid-air interface, leading to the eventual escape of the gas to the environment[2–4]. In everyday settings, rather than forming directly upon container walls, bubbles commonly form inside small cellulose fibers left behind during cleaning[5,6]. These cavities ('lumen') are remarkable sites for consistent and rapid bubble growth and release since a pocket of gas remains behind after each pinch-off event to seed the next growth. The coalescence of diffusively growing bubbles itself presents a classical modeling challenge[7,8], as is the process by which coalescing bubbles depart from a wall[9–12]. See in particular the reviews by Liger-Belair[13] and Lohse[14].

Supersaturated fluids also appear in geophysical settings. Explosive fragmentation of particulate matter in magma can lead to volcanic eruptions. A Strombolian eruption is caused by bubbles which coalesce into rising 'gas slugs', transporting gas and entraining magma flow towards the surface[15–17]. Such liquids also appear in industrial processes like deacidification and fractionation of oils[18,19], and in biological settings (e.g., blood and tissues during decompression)[20,21]. Large-scale flows associated with bubble motion depend on the geometry of the container, and can result in peculiar downward bubble motion and even bubble waves and cascades in fluids from stout beers to magma[22–24].

When a free body is introduced into a supersaturated environment it presents new sites at which bubbles can nucleate and grow - a field of such bubbles on a body surface can result in surprising dynamics. Using nothing more than carbonated water and raisins (Fig. 1), periodic vertical body motions can be observed for nearly two hours, though the time between excursions slows considerably after the first 20 min (see Movie S1). This phenomenon has earned numerous playful names by a variety of fizziologists[25], from dancing raisins to divers[26,27] and fizz balls[28]. Similar body oscillations can also be generated by reactions in chemical gardens[29,30]. Recently, this phenomenon has been investigated using peanuts and beer, based on a common practice in Argentinian drinking establishments[31]. Different contact angles of individual attached bubbles show the importance of surface

---

[1]Department of Mathematics, University of Wisconsin-Madison, Madison, WI 53706, USA. [2]Department of Chemical & Biological Engineering, University of Wisconsin-Madison, Madison, WI 53706, USA. ✉e-mail: spagnolie@math.wisc.edu

roughness on bubble formation[31]. Such effects add to a growing list of fundamental scientific findings which have emerged from the kitchen[32,33].

In this paper, we explore these oscillating dynamics using experiments and simulations and match numerous features to theoretical predictions. Experiments are used to measure the force development on a spherical body fixed in carbonated water, and its oscillatory dynamics when free. Using a discrete bubble model, and a continuum model which incorporates the dynamics of a surface buoyancy field, ranges of body and fluid properties are provided for which the system is most dynamic, and those for which body excursions are suppressed. Simulations are used to explore the dependence on system parameters in a more controlled setting. Body rotations are found to be critical for the onset of periodic vertical motions of large bodies, which in turn can produce body wobbling, rolling, and multiperiod surface return in a damped bouncing dynamics.

## Results

### Experiments

**Mass loss of a supersaturated fluid.** We first measured the change in gas concentration in a supersaturated fluid upon depressurization by examining the fluid's mass loss over time. An empty glass vessel with a square cross-section of side length $L = 8.9$ cm was filled with a just-opened can of Klarbrunn-brand carbonated water which was stored at

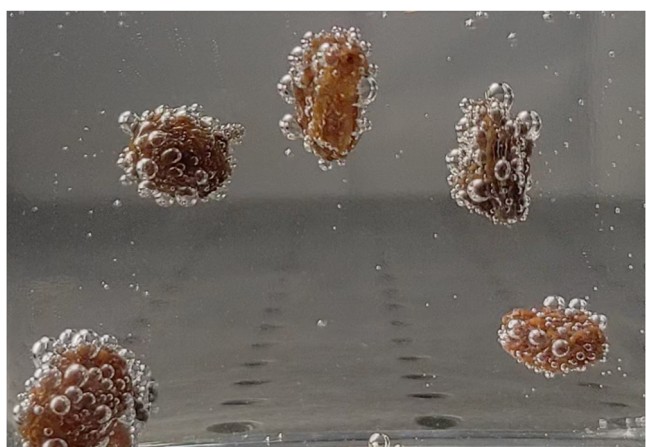

**Fig. 1 | Dancing raisins.** Raisins in carbonated water present numerous folds conducive to bubble nucleation and growth; these bubbles may then lift the body upward against gravity, only to release it upon arrival at the free surface (see Movie S1).

room temperature (21.6 °C). The fluid was poured into the vessel and then left alone for two hours at the same room temperature. Time, denoted by $t$, is measured in minutes, and $t = 0$ corresponds to the time just after depressurization (the moment when the can was opened). The mean mass loss across five experiments at each time is shown in Fig. 2a as a thin red curve. In each experiment the initial volume of fluid, $V_f$, was approximately 355 cm³ (12 oz); the total mass loss, $M_e(t)$, over the course of two hours was roughly 0.3% of the initial mass.

Rapid mass loss over the first 20 min was due to the growth and detachment of bubbles on the container surface, and diffusive transport of gas from the free surface, as discussed below. The remaining 100 min revealed linear behavior due to water evaporation[34]. At long times we observed that $dM_e/dt \approx k_e$ for relatively large $t$, where $k_e = 4 \times 10^{-3}$ g/min (the same value was found using only tap water). The solid blue curve in Fig. 2a shows $M_e(t) - k_e t$, expected to be the mass loss of $CO_2$, and the standard deviation at each time across experiments is shown using error bars.

Denoting the volume-averaged gas concentration at time $t$ by $\bar{c}(t)$, with units of g/cm³, the mass of $CO_2$ in the vessel is given by $\bar{c}(t)V_f$ (an adjustment due to evaporative volume change is negligible). The mass loss from the empty vessel is written as $M_e(t) = V_f(\bar{c}(0) - \bar{c}(t)) + k_e t$, where $\bar{c}(0)$ is the initial concentration of $CO_2$. The supersaturation ratio, $S(t) = (\bar{c}(t)/c_s - 1)$, measures the gas concentration level compared to a critical value $c_s$, an equilibrium concentration corresponding to a partial pressure of gaseous $CO_2$ at 1 atm[13,35]. At room temperature in water this value is $c_s = 1.48$ g/L[36,37]. The gas concentration decreases until $S(t)$ reaches a value below which bubbles no longer form on or near the container walls; in this case, most likely in the lumen of adhered fibrous matter[5,6]. We denote this minimum value associated with the container as $S_{mc}$ and discuss its genesis more fully in §II B. The difference $S(t) - S_{mc}$ may then be inferred from the mass loss data, and is plotted in Fig. 2b. The dynamics of $S$ are modeled by a Riccati equation, $\dot{S} = -(S - S_{mc})/T_r - q(S - S_{mc})^2$ with the dot denoting a time derivative, for reasons to be discussed, which results in a predicted evolution of the form

$$S(t) = S_{mc} + \frac{(S_0 - S_{mc})\exp(-t/T_r)}{1 + \chi(1 - \exp(-t/T_r))}, \tag{1}$$

where $S_0 = S(0)$ is the supersaturation ratio at $t = 0$, and $\chi = qT_r(S_0 - S_{mc})$. This function is plotted atop the experimental measurements in Fig. 2b as a dashed curve using fitted parameters $S_0 - S_{mc} = 1.66$, $\chi = 2.5$, and $T_r = 36.2$ min. With $S_0 - S_{mc} \approx S_0$, the observations here are in line with carbonation levels reported in other sparkling beverages[4,38], particularly given the substantial gas loss while pouring[4,39].

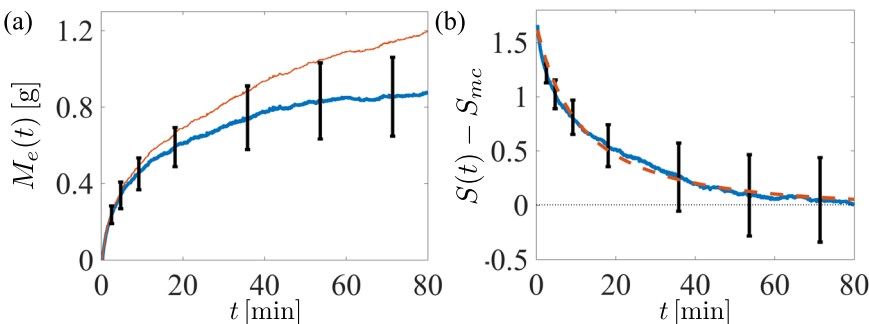

**Fig. 2 | Gas loss from an empty container of carbonated water. a** The mass loss in grams of one can of carbonated water during the first 80 min after depressurization (thin red line) at time $t$ averaged over four independent experiments, and the mass loss after subtracting a constant linear loss due to evaporation (dark blue line).

Error bars show the standard deviation across experiments at each time. **b** The $CO_2$ gas supersaturation ratio, minus a stationary minimum value below which bubbles do not form on the container, $S(t) - S_{mc}$, inferred from the data in (**a**) (dark blue line). The dashed line is a best-fit curve using (1).

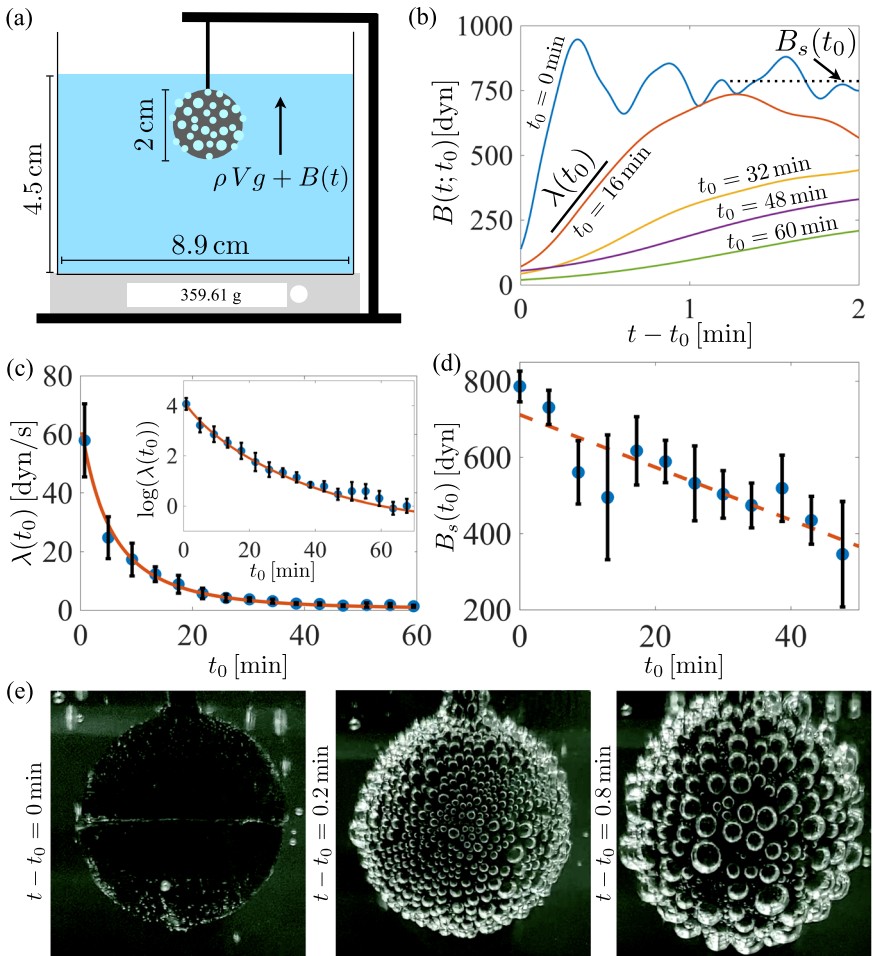

**Fig. 3 | Measuring the surface buoyancy on a fixed body. a** Experimental setup for measuring the surface buoyancy. A spherical 3D-printed body is held fixed in place in carbonated water for 4 min, then removed from and reinserted into the fluid. Bubbles form, grow, merge and detach, and the force is registered by the scale. **b** The surface buoyancy force, $B(t; t_0)$ during a single experiment. The force increases at a rate $\lambda(t_0)$ which depends on the insertion time $t_0$, tapering off to a value $B_s(t_0)$ (measured later, at $t_0 + 4$ min). The rate $\lambda(t_0)$ is computed using the lines shown. **c** The growth rate, $\lambda(t_0)$. The mean and standard deviation across five experiments are shown as symbols and error bars. The solid curve is based on a model which assumes $\lambda \propto S^{3/2}$ and using (1). **d** The stabilized surface buoyancy force, $B_s$, measured at $t_0 + 4$ min, along with a linear fit. **e** Bubble configuration on the spherical surface at three different moments after an early insertion time of $t_0 = 1$ min.

**Surface buoyancy growth with a fixed body.** Next, a body was held fixed in the fluid, and we measured the force development due to bubble growth on its surface. A sphere of radius 1 cm composed of Polylactic acid (PLA) was printed with an Ultimaker 3+ 3D printer using a 0.15 mm nozzle. A schematic of the experimental setup is shown in Fig. 3a. The fluid was depressurized and poured gently into the container, and placed upon the digital scale. The test sphere was then lowered into the carbonated water at different insertion times, denoted by $t_0$, and affixed to the table below. As bubbles grew on the surface (see Fig. 3e) they imposed an upward vertical force on the stationary sphere, and the opposing downward force was registered by the scale. The sphere was briefly removed from the fluid every 4 min, then reinserted while still wet. Videos of bubble growth, coalescence, and arrival at a fluctuating steady state at insertion times $t_0 = 1$ min and $t_0 = 10$ min are included as Movies S2–S3.

The scale-registered weight of the system, denoted by $F(t)$, was decomposed as $F(t) = F(0) + M_e(t)g + B(t; t_0)$. $F(0)$ includes a constant buoyancy force due to the volume of the displaced fluid; $M_e(t)$ is the mass change of a body-free fluid previously described; and $B(t; t_0)$ is the surface buoyancy force, the contribution to the buoyancy by the bubbles since the insertion time, $t_0$. The weight loss due to gas escape from an empty vessel is small, but it is of comparable magnitude to the

force on the body for roughly the first 10 min, and must be accounted for here. $F(t) - F(0)$ was measured directly, which then provided an indirect measurement of the added buoyancy force, $B(t; t_0)$.

Figure 3b shows the surface buoyancy force as a function of time since insertion, $t - t_0$, for insertion times up to 60 min. Upon insertion the force increased, at a rate which diminished as the insertion time became larger and the fluid became calmer. As discussed in §II B, the radius $a(t)$ of an individual bubble is expected to grow roughly as $a(t) \approx (2DS(t_0)(t - t_0))^{1/2}$, where $D$ is the diffusion constant for $CO_2$ at room temperature. The associated volumetric growth suggests a total force on the body proportional to $(t - t_0)^{3/2}$; however, once heterogeneous bubble coalescence begins the growth appears closer to linear in time (seen for insertion times less than 30 min in Fig. 3b). This growth rate, denoted by $\lambda(t_0)$, is shown in Fig. 3c. The solid curve corresponds to a prediction $\lambda(t_0) = kS(t_0)^{3/2}$, where $k$ is a proportionality constant, revealing an excellent match to the experimentally measured growth rates. Equivalently, defining $\lambda_0$ to be the linear growth rate at $t = 0$, we may write

$$\lambda(t_0) = \lambda_0 \left( \frac{S(t_0)}{S_0} \right)^{3/2} = \lambda_0 \left( \frac{(S(t_0) - S_{mc}) + S_{mc}}{(S_0 - S_{mc}) + S_{mc}} \right)^{3/2}. \quad (2)$$

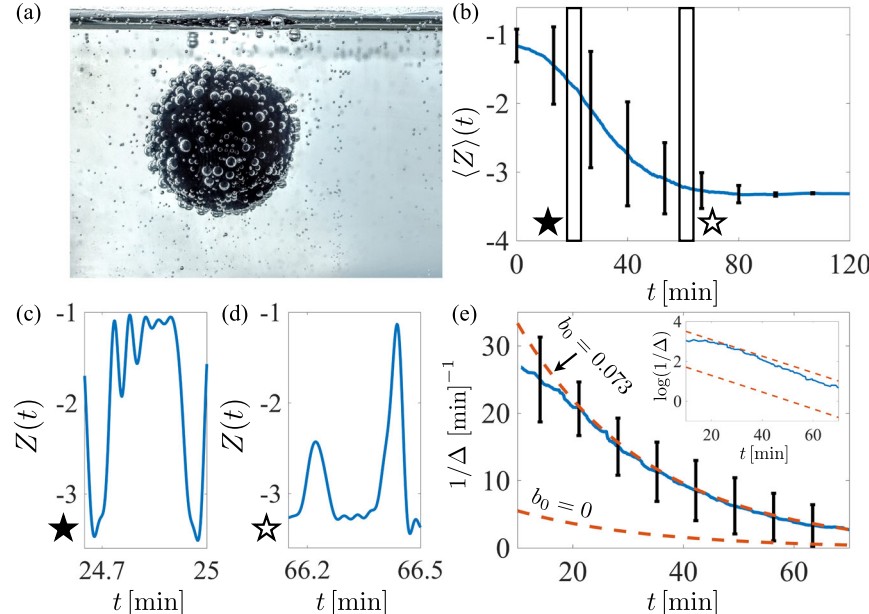

**Fig. 4 | Long-time dynamics of a free body. a** A freely moving 3D-printed sphere of radius 1 cm and mass 4.25 g in carbonated water, rotates and returns to the free surface. **b** The locally averaged position over a window of ± 5 min for a single experiment is shown as a dark curve; the standard deviation is shown with error bars. **c** The instantaneous vertical position from (**b**). The body tends to visit the surface numerous times in succession, clearing off more and more of the surface, before plummeting. **d** At later times, body rising events are often cut short by premature detachment of large individual bubbles. **e** The dancing frequency, $f := 1/\Delta(t)$, where $\Delta(t)$ measures the averaged excursion time (over a ± 5 min window) from surface-departure to surface return. The solid curve shows the mean over three experiments, error bars show the standard deviation across experiments, and the dashed lines are from (11). Inset: the same, on a log-linear scale.

Using $S(t) - S_{mc}$ from the previous experiment with no immersed body (see Fig. 2b), the model best fits the data on a logarithmic scale using $\lambda_0 = 58.4$ dyn/s and $S_{mc} = 0.020$. By comparing with the previous experiment, we find $S_0 \approx 1.68$.

The surface buoyancy force eventually stabilized to a value $B_s(t_0)$ measured four minutes after reinsertion time $t_0$. This value is shown in Fig. 3d along with a linear fit, $B_s(t_0) \approx 712$ dyn-(6.9 dyn/min)$t$. The saturated value $B_s(t_0)$ diminished far more slowly than did the growth rate $\lambda(t_0)$ - roughly, at later insertion times, bubbles merely take longer to accumulate and grow until reaching a critical size, at which time they pinch off and depart alone. Whether the slow decay was due to surface wetting or other phenomena remains unclear. A simple method for estimating $B_s$ is included in the Supplementary Information.

After the buoyancy force stabilized, large fluctuations were observed. They were most apparent at smaller insertion times due to large bubble surface sliding and detachment events, which carry numerous other bubbles away at the same time. This surface cleaning effect has recently seen more specific attention[12]. Detachment is expected beyond a critical 'Fritz radius' where buoyancy overwhelms capillary forces[31,40,41]. Bubbles have other opportunities to depart during merging events via self-propelled detachment[9].

The same force development measurement was performed using a skewer of 8 Sunmaid raisins, as reported in the Supplementary Information. The initial maximum force per raisin due to bubbles was $B_s \approx 100$ dyn, and the initial growth rate per raisin was $\lambda_0 \approx 20$ dyn/s.

**Levitation and dynamics of a freely moving sphere.** In a final set of experiments, the printed sphere was placed into the fluid, free to move (Fig. 4a). The body, with radius $A = 1$ cm and mass $m = 4.25$ g, was inserted into the fluid just after depressurization and pouring, and its motion was recorded with a Nikon D7000 DSLR camera with a 24 fps framerate for two hours (see Movies S4–S6). The body's vertical position at each time, denoted by $z = A Z(t)$ (with the fluid surface at $z = 0$), was recovered using an image tracking code written in Matlab. The experiment was performed five times.

A slow decay of the highly oscillatory dynamics is observed upon locally averaging the vertical position. Figure 4b shows the mean vertical position at time $t$ over a window $[t - 5$ min, $t + 5$ min] during one experimental run. The free surface is located at $z = 0$, and the body is just touching the free surface when $Z = -1$. For the first 5 min the body spends most of its time near the free surface-bubbles cleaned from the body are rapidly replaced. Eventually, the body begins to spend more time in the bulk fluid, and then on the container floor, where the body resides for longer and longer 'charging' times before rising again.

The instantaneous vertical position starting at $t = 24.7$ min is shown in Fig. 4c, revealing a type of damped 'bouncing' dynamics from the free surface. Multiple visits to the fluid-air interface in succession are commonly observed; each visit clears different portions of the body surface, resulting eventually in a longer excursion into the bulk. Figure 4d shows common late-stage behavior: long periods on the container floor, punctuated by increasingly rare rising events. Rising often ends prematurely due to the detachment and departure of single large bubbles. The late-stage surface forces tend to be dominated by such individuals, and the body's fate can be dictated by their singular activity.

An excursion is defined as having occurred if the body's center of mass drops below one diameter beneath the free surface before returning upward and crossing the same vertical threshold. The times at which the $j$th excursion begins and ends are denoted respectively by $t_j$ ($Z(t_j) = -2$ and $Z'(t_j) < 0$), and $p_j$ ($Z(p_j) = -2$ and $Z'(p_j) > 0$). The $j$th excursion time is given by $\Delta_j := p_j - t_j$, for $j \geq 1$. The body undergoes roughly 300–700 excursions during the two-hour experiment. A smoothed excursion time, $\Delta(t)$, is defined as another moving average using ± 5 min on either side of time $t$. The dancing frequency, $f := 1/\Delta$, is shown in Fig. 4e, and on a log-linear scale in the inset, including error bars representing one standard deviation at that time across three independent experiments. A clear monotonic decrease in the frequency is observed with nearly exponential decay over time. The dashed curve corresponds to a theoretical prediction discussed below.

For bodies with a dense surface bubble coverage, body rotation at the surface plays a critical role. Early experiments, not shown here, constrained the motion to translation along and rotation about the vertical axis only. This dramatically inhibited surface bubble removal and associated dancing. Only a rare event of a large bubble leaving the surface could result in an excursion. Rather, if the body is free to rotate, then upon reaching the surface and losing a portion of the surface-bound bubbles, the body becomes unstable to rotation due to the remaining bubbles on the underbelly of the sphere. Once the body starts to rotate, the body is cleared of a larger number of buoyancy-conferring bubbles, and an excursion becomes far more likely (see Movie S4). The previously noted bouncing at the free surface is an additional consequence. Fluctuations from the active fluid, or other immersed bodies (Movie S5), which nudge the rotational instability to develop appear to be important as well. For bodies like raisins, whose trajectories are influenced more by individual bubble growth, lift, and removal, vertical dancing may be observed without need for such rotations.

### Mathematical model

**Gas escape.** There are a number of mechanisms by which gas may escape from the system: formation of bubbles on the container walls and on the immersed body, which eventually exits into the surrounding environment, and by diffusive transport through the fluid-air interface[13]. For the first mechanism, assuming simply that the frequency of surface bubble growth and bubble volume upon exit each to be linear in the supersaturation ratio, the gas loss from this process follows $\dot{S} = -q(S - S_{mc})^2$, where $q$ is a constant which encapsulates the number of bubble growth sites and their geometrical features, and $S_{mc}$ is the minimum value below which bubbles no longer form on or near the container walls (discussed below). This quadratic law is supported by the data in Fig. 2, and it suggests that this mechanism of gas loss is dominant for the first one or two minutes of the experiment. Gas loss due to growth on the immersed body and direct delivery to the surface may be neglected (see Supplementary Information).

The steady stream of bubbles rising from (near) the container walls drives a large-scale circulation flow on the scale of the container's lateral measure akin to intrinsic convection in particle sedimentation[39,42,43]. Such a flow is expected to continually replenish the region near the fluid-air interface with the well-mixed concentration from the bulk fluid. Balancing advection and diffusion near the free surface, the concentration is predicted to decay from its volume-averaged value inside the fluid to approximately zero outside the fluid across a small boundary layer of size $\delta = \sqrt{DH/(2U_b)}$, where $H$ is the fluid depth and $U_b$ is the velocity of bubble rise (see Supplementary Information). This motivates a model for the supersaturation ratio, $\dot{S} = -(S - S_{mc})/T_r$, where $T_r = V_f \delta/(DS_f)$ is a relaxation timescale, with $S_f$ the free surface area and $V_f$ the fluid volume. Using the dimensions of the experiment, $L = 8.9$ cm and $H = 4.5$ cm, the diffusion constant for $CO_2$ in water $D = 1.85 \times 10^{-5}$ cm²/s[44], and an observed bubble rise velocity of $U_b \approx 1$ cm/s, this gives $\delta = 65$ μm. Then with $S_f = 79$ cm², and fluid volume $V_f = 355$ cm³, the predicted relaxation time $T_r$ is 26 min, within range of the best-fit value used in Fig. 2. Combining the mechanisms above, $\dot{S} = -(S - S_{mc})/T_r - q(S - S_{mc})^2$, produces the expression in (1).

**Discrete and continuum buoyancy growth models.** We consider two models of bubble/buoyancy growth, a discrete model and a continuum model, as each can be more appropriate depending on the body size and surface properties. In the discrete model, each of $N$ bubbles are assumed to grow independently according to the bubble growth law attributed to Scriven, which builds upon the Rayleigh-Plesset equation:

for an isolated bubble of radius $a(t)$, we have

$$\dot{a} = \frac{D}{a}\left(S - \frac{2\sigma}{pa}\right), \tag{3}$$

where $D$ is a diffusion constant ($D = 1.85 \times 10^{-5}$ cm²/s for $CO_2$ in water[44]), $S(t)$ is the supersaturation ratio, $\sigma$ is the surface tension of water ($\sigma \approx 70$ mN/m = 70 dyn/cm at room temperature), and $p$ is the pressure near the bubble ($p \approx 1$ atm = $10^6$ dyn/cm²)[1,45–47]. The force conferred to the body is the buoyancy experienced by the bubble. In the discrete model, a bubble is removed from the surface when its position on the body exits the fluid, or when it reaches a critical size for pinch-off, $a_p$, at which time it is immediately rebirthed (once in the fluid) with size $a_0$, a characteristic scale of surface roughness.

Of particular importance at longer times, there is a supersaturation ratio below which bubbles are overwhelmed by pressure and surface tension and cease to form on the body, $S = 2\sigma/pa_0$. The initial bubble size, $a_0$, depends upon the surface and the nature of bubble nucleation there. The length scale of surface roughness on the body may be associated with such initial bubble sizes; the lumen diameter of the fibers left behind during cleaning on the container walls is another. This value for the bubble formation near the surface was denoted $S_{mc}$, and inferred from the experiments in §II to be $S_{mc} \approx 0.020$. This suggests that the surface roughness (or remnant material) scale on the container wall is $2\sigma/(pS_{mc}) \approx 30$ μm, which matches the diameter of cellulose fibers[5].

On the body, meanwhile, using the resolution of the 3D printer of 0.15 mm to estimate $a_0 \approx 0.015$ cm, a minimal supersaturation ratio for bubble growth on the body, denoted by $S_{mb}$, may be roughly 0.010. That $S_{mb} < S_{mc}$ indicates that bubbles should continue to form on the body after they cease to form on/near the container walls. Indeed, after a carbonated fluid was left alone for hours until it appeared 'flat' (bubble-free), inserting at that time a 3D-printed body or a raisin resulted in bubble growth on the body and the onset of body rising events. A detailed determination of $S_{mb}$ generally involves not only the surface roughness but also its chemistry, as the contact angle has been shown to be important for bubble nucleation[31].

The second model considered is a continuum model, more appropriate when the body is covered in a large number of bubbles which are continually growing, merging, and detaching. With the local maximum of the added traction given by $B_s/(4\pi A^2)\hat{\mathbf{z}}$, the instantaneous portion of this local contribution is defined as $bB_s/(4\pi A^2)\hat{\mathbf{z}}$, where $b(\mathbf{x}, t) \in [0, 1]$ and $\mathbf{x}$ is a point on the body surface. The evolution of this buoyancy fraction is then modeled as

$$\frac{d}{dt}b(\mathbf{x}, t) = \frac{\lambda(t)}{B_s}(1 - b(\mathbf{x}, t)), \tag{4}$$

which holds pointwise at every position $\mathbf{x}$ on the body surface. Spatial variations in $b(\mathbf{x}, t)$ are possible, and generically produce a body torque. Holding the supersaturation ratio fixed and neglecting surface tension and pressure, the bubble growth law suggests a bubble radius growth $a(t) \sim (a_0^2 + 2DS(t_0)(t - t_0))^{1/2}$. Since the contributed buoyancy is proportional to $a(t)^3$, this suggests a growth rate $\lambda(t)$ which scales as $S^{3/2}$ for appreciable times, consistent with the measured data. This leads to the model for the growth rate given in (2).

In both models, the instantaneous lifting force and torque in the lab frame are written as $B_s F_B[b]\hat{\mathbf{z}}$ and $AB_s \mathbf{L}_B[b]$, respectively. In the continuum model, $F_B[b] = \langle b \rangle := (4\pi)^{-1}\int_{S_0} b(\mathbf{X}, t)\, dS_0$, where $dS_0$ is the surface area element in the body frame, and $\mathbf{L}_B[b] = (\mathbf{Q}\langle b\mathbf{X} \rangle) \times \hat{\mathbf{z}}$. Or, defining the center of surface buoyancy in the body frame as $\mathbf{M} = \langle b\mathbf{X} \rangle = (4\pi)^{-1}\int_{S_0} \mathbf{X}b(\mathbf{X}, t)\, dS_0$ and $\mathbf{m} = \mathbf{QM}$ the same in the lab

frame, we may write $\mathbf{L}_B[b] = (\mathbf{QM}) \times \hat{\mathbf{z}}$. Additional details are given in the Supplementary Information.

**Equations of motion and dimensionless groups.** Taking $T := \sqrt{A/g}$ to be a characteristic time and $A/T = \sqrt{Ag}$ to be a characteristic speed, we write the position of the body centroid as $\mathbf{r}(t) = A \cdot Z(t)\hat{\mathbf{z}}$, the vertical velocity as $(A/T)W$, and the body rotation rate as $T^{-1}\mathbf{\Omega}$. Force and torque balance, with the dimensionless time $s := t/T$, are expressed as

$$\frac{dW}{ds} = -1 + \frac{\alpha(s)}{\mathcal{M}} + \beta F_B[b] - \frac{9 C_T}{2\mathcal{M}\text{Re}} W, \qquad (5)$$

$$\frac{d}{ds}(I_R \mathbf{\Omega}) = \beta \mathbf{L}_B[b] - \frac{6 C_R}{\mathcal{M}\text{Re}} \mathbf{\Omega}, \qquad (6)$$

where we have introduced the following dimensionless numbers,

$$\mathcal{M} = \frac{m}{\rho V}, \ \ \beta = \frac{B_s}{mg}, \ \ \Lambda = \frac{(A/g)^{1/2}\lambda_0}{B_s}, \ \ \tau = \frac{T_r}{(A/g)^{1/2}}. \qquad (7)$$

Here, $V$ is the body volume, $V_S(s)$ is the submerged volume at time $s$, $\alpha(s) := V_S(s)/V \in [0,1]$, and $mA^2 I_R$ is the moment of inertia, where $I_R = 2/5$ for a rigid sphere. Hydrodynamic drag and torque coefficients, respectively, are given by $C_T = 1 + .0183\,\text{Re}\,|W|$, and $C_R = 1 + 0.0044\sqrt{\text{Re}\,|\mathbf{\Omega}|}$, with $\text{Re} = \rho A^{3/2} g^{1/2}/\mu$ the Reynolds number (The Reynolds number is defined as $\text{Re} = \rho UA/\mu$, with $\rho$ the fluid density, $U$ a characteristic speed, $A$ a characteristic length scale, and $\mu$ the fluid viscosity, and gives a measure of the importance of inertia relative to viscous dissipation.). Bubbles generally affect the drag on the body but we neglect this detail here. The surface buoyancy fraction $b$ evolves at each point on the body as

$$\frac{d}{ds}b(\mathbf{x},s) = \Lambda g(s)(1 - b(\mathbf{x},s)), \qquad (8)$$

where $g(s) = (S(Ts)/S_0)^{3/2}$, and using Eqn. (1),

$$g(s) = \left(\frac{S_{mc}}{S_0} + \frac{(1 - S_{mc}/S_0)\exp(-s/\tau)}{1 + \chi(1 - \exp(-s/\tau))}\right)^{3/2}. \qquad (9)$$

Any points $\mathbf{x}$ on the body surface which are outside of the fluid are given the value $b = 0$; only once those points reenter the fluid does the growth there begin again according to (8). The system is closed by tracking the body's position and orientation: $\dot{Z} = W$ and $\dot{\mathbf{Q}} = \mathbf{Q}\,\hat{\Omega}$ where dots denote derivatives upon the dimensionless time $s$, and $\hat{\Omega}\mathbf{q} := \mathbf{\Omega} \times \mathbf{q}$ for any vector $\mathbf{q}$. Initial conditions are generally taken to be $Z(0) = -1$, $W(0) = 0$, $\mathbf{\Omega}(0) = \mathbf{0}$, $\mathbf{Q}(0) = \mathbf{I}$, and $b(\mathbf{x}, 0) = 0$.

The system is thus characterized by a mass ratio, $\mathcal{M}$, the relative lifting force $\beta$, which we term the fizzy lifting number ("Fizzy lifting drinks! They fill you with bubbles, and the bubbles are full of a special kind of gas, and this gas is so terrifically lifting that it lifts you right off the ground just like a balloon, and up you go until you're bumping against the ceiling!" -Charlie and the Chocolate Factory, by Roald Dahl.), initial bubble growth rate, $\Lambda$, and relaxation time, $\tau$, the minimal supersaturation ratio for bubble growth along the container, $S_{mc}$, the initial supersaturation ratio, $S_0$, and the Reynolds number. For the 3D-printed body used above, using $A = 1$ cm, $m = 4.25$ g, $B_s = 700$ dyn, $\lambda_0 = 58.4$ dyn/s, and $T_r = 36.2$ min, and for water $\rho = 1$ g/cm³ and $\mu = 0.01$ g/(cm s), we find $(\mathcal{M}, \beta, \Lambda, \tau, \text{Re}) = (1.015, 0.17, 2.6 \cdot 10^{-3}, 6.8 \cdot 10^4, 3.1 \cdot 10^3)$. For a raisin, using a prolate spheroidal body with semi-major axis length $A = 0.6$ cm and semi-minor axis lengths 0.4 cm, mass $m = 0.45$ g, $B_s = 100$ dyn, and $\lambda_0 = 20$ dyn/s, we find $(\mathcal{M}, \beta, \Lambda, \text{Re}) = (1.12, 0.23, 4.9 \cdot 10^{-3}, 1.5 \cdot 10^3)$. These values of $\beta$ and $\Lambda$ make raisins particularly strong dancers.

The body is positively buoyant and floats without bubbles if $\mathcal{M} < 1$. If $\mathcal{M} > 1$, the body can only be lifted upward against gravity if the fizzy lifting number $\beta$ is sufficiently large; namely $\beta + \mathcal{M}^{-1} - 1$ must be positive. This gives a range of masses for which oscillatory dynamics are expected to reside: $\mathcal{M} \in (1, 1/(1 - \beta))$ if $\beta < 1$, and $\mathcal{M} \in (1, \infty)$ if $\beta \geq 1$. The 3D-printed bodies are predicted to dance with density ratios $\rho_s/\rho \in (1, 1.20)$; raisins are expected to dance in a similar range, $\rho_s/\rho \in (1, 1.29)$. With $B_s$ generally scaling with the surface area and $m$ scaling with the body volume, $\beta$ is generically larger for very small bodies, and bubble-induced lifting is expected for such bodies to be more immediate.

**Dancing frequency.** We approximate the solution to (8) by neglecting the early period of rapid gas escape and assuming $S_{mc} \ll S_0$. Taking $g(s) \approx (\exp(-s/\tau)/(1 + \chi))^{3/2}$, with $b(\mathbf{x}, s_0) = b_0$, we find

$$b(\mathbf{x}, s) = 1 - (1 - b_0)\exp\left(\frac{2\Lambda\tau}{3(1+\chi)^{3/2}}\left[e^{-3s/2\tau} - e^{-3s_0/2\tau}\right]\right). \qquad (10)$$

The dimensionless 'charging time' before the total buoyancy overcomes gravity is given by $s_{charge} := s_c - s_0$ such that $\beta\langle b\rangle(s_c) + 1/\mathcal{M} - 1 = 0$, which from (10) is given by

$$s_{charge} = \frac{2\tau}{3}\log\left(\frac{1}{e^{-3s_0/2\tau} - \frac{3(1+\chi)^{3/2}}{2\Lambda\tau}L_\beta}\right) - s_0, \qquad (11)$$

where $L_\beta = \log\left(\frac{\beta\mathcal{M}(1 - b_0)}{1 - (1 - \beta)\mathcal{M}}\right)$. For a sufficiently small container the charging time serves as a proxy for the excursion time, with transit from one surface to another playing only a small role. Two curves corresponding to $f = 1/\Delta \approx 1/t_{charge}$ (with $t_{charge} := Ts_{charge}$) are included in Fig. 3e, one with $b_0 = 0$ and one which best fits the data with $b_0 = 0.073$. The frequency is sensitive to $b_0$, which points yet again to the importance of rotations, and how many bubbles are removed upon each surface visit.

For insertion times $s_0 > s_{fun}$, where $s_{fun} = (2\tau/3)\log(2\Lambda\tau/(3(1+\chi)L_\beta))$, the charging time in (11) is infinite. Using the values from the experiments the associated dimensionless time is $t_{fun} := Ts_{fun} = 170$ min. A separate approximation starting from (8) is more appropriate at long times, assuming $S_{mc} > S_{mb}$ (i.e., bubbles continue to form on the body after they cease to form along the container walls). In this case, as $s_0/\tau \to \infty$, we have $g(s_0) \sim (S_{mc}/S_0)^{3/2} =: g_\infty$, leading to a constant charging time of $s_{charge} = (\Lambda g_\infty)^{-1}L_\beta$. Using the experimental parameters this gives a final dimensional charging time of between 14 min and 2.5 min, for initial coverages $b_0 = 0$ and $b_0 = 0.073$, respectively. Once $S \leq S_{mc}$, bubbles no longer form along the container and the primary mechanism driving gas escape is removed. The body can continue to form bubbles and perform its low-frequency dance, even in an otherwise quiescent fluid. Since we neglect the days-long timescale of pure diffusive transport, and convective diffusion affected by the body motion, this low-frequency dancing is predicted to carry on indefinitely. Dancing with a mean frequency of 1.5 min⁻¹ was indeed observed in one experimental run for the last 4 h of a 5 h run.

**Simulations.** To examine the dynamics in a more controlled environment we solve (5–8) numerically. Figure 5 shows the dynamics of a body with bubbles at the vertices of a regular icosahedron ($N = 12$), using the discrete bubble model, with $(\mathcal{M}, \beta, \Lambda, \text{Re}) = (1.02, 0.17, 0.003, 3100)$. The rebirth and pinch-off radii used are $a_0/A = 0.03$ and $a_p/A = 0.24$, respectively. The bubbles first grow to sufficient stature to lift the body to the surface, where the topmost bubbles are released. The body begins to descend slowly before undergoing a rotation, returning soon after to the surface. Each visit to the surface includes bubble removal, a body rotation, additional bubble removal, and then a plummeting to deeper waters.

Figure 6a shows the dimensionless vertical dancing frequency, $Tf$, using the discrete bubble model as a function of the number of bubbles, all placed at the vertices of a regular polyhedron, all with $(\mathcal{M}, \beta, \Lambda) = (1.015, 0.17, 0.0016)$. To isolate the role of bubble position we adjust the maximum bubble size $a_p/A = (3B_s/(4\pi \rho g N))^{1/3}$ so that each body achieves the same maximal surface buoyancy ($\beta$) if all bubbles are at their maximal (pinch-off) size. The frequency increases with $N$, as bodies with a greater number of bubbles can begin their descent while still maintaining partial surface coverage, and less bubble growth is needed before the body becomes positively buoyant again.

To further explore body rotations, Fig. 6b shows the mean body rotation rate as a function of the growth rate, $\Lambda$, using both the discrete bubble model (shown as symbols) and the continuum model (as a solid curve). With increasing $\Lambda$ the body spends more time at the surface, and experiences opportunities to rotate more frequently, if not more rapidly. The growth is approximately logarithmic in $\Lambda$, and for small $\Lambda$ there is close agreement between the discrete and continuum models.

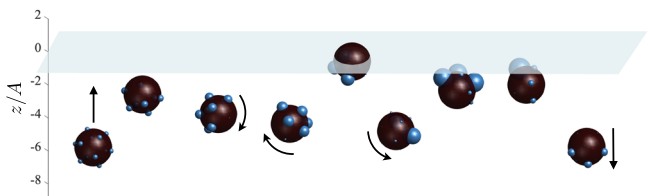

**Fig. 5 | Simulations using the discrete bubble model.** A body with $(\mathcal{M}, \beta, \Lambda, \mathrm{Re}) = (1.02, 0.17, 0.003, 3100)$, and bubbles at the vertices of an inscribed icosahedron ($N = 12$), undergoes "bouncing" dynamics near the surface. After the bubbles nearest to the surface are removed, the center of surface buoyancy rests below the center of mass, resulting in a torque and eventual body rotation. Only after a few returns to the surface to clear off more bubbles does the body begin a large excursion back towards the container floor. See Movie S7.

The discontinuity in the discrete model is due to the onset of premature bubble detachment at large bubble growth rates (see Movie S7). For sufficiently large growth rates ($\Lambda > 0.008$) the bubbles grow from their initial size $a_0$ to the pinch-off size $a_p$ on a shorter timescale than the body's excursion time. Consequently, most, if not all, bubbles are removed near the time that the body reaches the surface, nearly eliminating the torques on the body through pinch-off alone and thus dampening rotations.

Figure 6c shows the mean rotation rate instead as a function of $\beta$ for a selection of growth rates, $\Lambda$, using the discrete bubble model, while Fig. 6d shows the same using the continuum model. Generally, larger values of $\beta$ are associated with larger torques, and thus faster body spinning. The discrete model shows again the importance of premature bubble pinch-off and departure at large bubble growth rates. For $\Lambda = 0.016$, if $\beta$ is small, it is common for most of the bubbles to pinch off before the body traverses the full length of the container. For large $\beta$, the body emerges completely out of the fluid in a dramatic jump, and all bubbles are removed leaving none to generate a torque.

**Wobbling and rolling.** For bodies which are large relative to the maximum bubble size, body rotations are commonly observed, as are a related dynamics: wobbling. As a coarse approximation we consider bubbles to have been removed from one half of the spherical surface, which then regrow with rate $\Lambda$. Writing the center of surface buoyancy in the body frame as $\mathbf{M} = \mathbf{M}(0) \exp(-\Lambda s)$, with $\mathbf{M}(0) = (4\pi)^{-1} \langle b\mathbf{X}\rangle = (1/4)\hat{\mathbf{z}}$, then with $\mathbf{q}_1 = \cos(\theta)\hat{\mathbf{x}} + \sin(\theta)\hat{\mathbf{z}}$ and $\mathbf{q}_2 = \hat{\mathbf{y}}$, the (dimensionless) torque in the lab frame is $\mathbf{L}_B = -(1/4)\exp(-\Lambda s)\sin(\theta)\hat{\mathbf{z}}$. Writing $\boldsymbol{\Omega} = \dot{\theta}\hat{\mathbf{y}}$, and neglecting the nonlinear part of the hydrodynamic moment, we arrive at $\ddot{\theta} = -a_1(s)\sin(\theta) - a_2\dot{\theta}$, where $a_1(s) = 5\beta \exp(-\Lambda s)/8$ and $a_2 = 15/(\mathcal{M}\,\mathrm{Re})$, the equation for a damped nonlinear oscillator with diminishing torque.

Wobbling diminishes either by viscous damping or by bubble (re) growth. A characteristic initial ($s = 0$) wobbling frequency from the above is $f_{wobble} \approx \sqrt{a_1}/(2\pi) \approx (5\beta/8)^{1/2}/(2\pi)$. For the 3D-printed body,

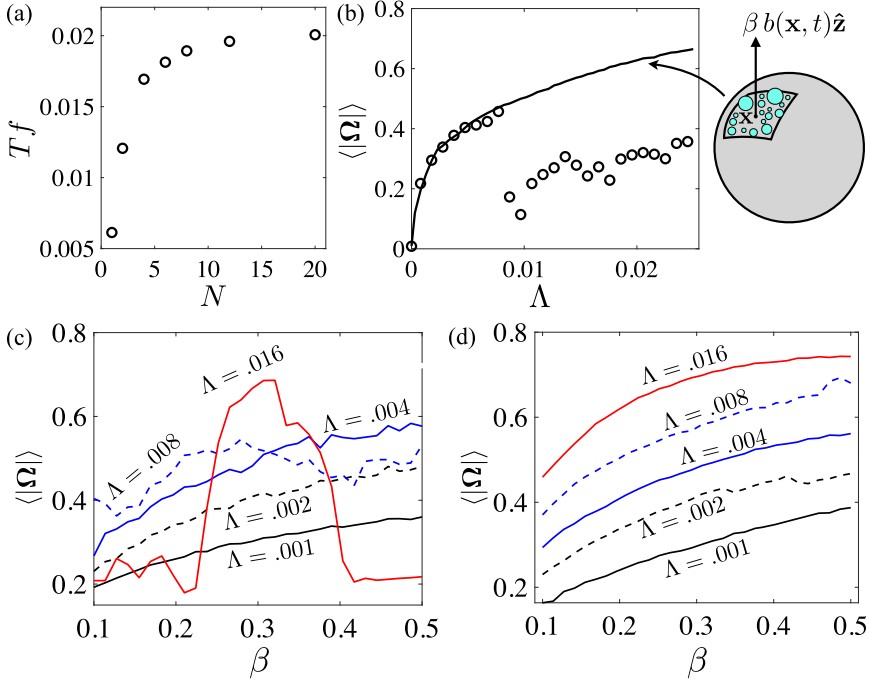

**Fig. 6 | Numerical simulations using the discrete and continuum models. a** The (dimensionless) dancing frequency, which increases with the number of bubbles, $N$, for fixed maximal surface lifting force ($\beta = 0.017$). **b** The mean rotation rate increases with the growth rate, $\Lambda$, in the continuum model (solid line). For the discrete model with $N = 12$ bubbles (circles), the mean rotation rate increases with $\Lambda$ until premature bubble detachment becomes important. **c** The mean rotation rate as a function of $\beta$, for a selection of growth rates $\Lambda$, using the discrete model with $N = 12$ bubbles. Rotations are damped at large $\beta$ and $\Lambda$ due to the body emerging from the fluid. **d** Same as (**c**) but using the continuum model.

using $\beta = 0.17$, the initial frequency is roughly $f_{wobble}/T \approx 1.6$ Hz. A raisin, meanwhile, due to its larger value of $\beta$, oscillates with a higher frequency of just over 2.4 Hz. These values are consistent with the experimental observations (see Movies S1, S4, and S5). This effect is similar to that seen in Quincke rotor dynamics, where surface charging is driven by electrohydrodynamics[48–54].

A transient rolling mode was also observed. Bubbles on the surface of a rolling body begin to grow upon reentry and are larger just before they exit, producing a sustained rolling torque. For this we consider a cylindrical body and a two-dimensional cross-section. If the body is fixed at a vertical position $Z = -\cos(\theta^*)$, with $\theta^* \in [-\pi, \pi]$, then the steady state lifting distribution assuming $\dot{b} = \Lambda$ (and using $\dot{b} = \Omega b_\theta$) is $b(\theta) = \Lambda\Omega^{-1}(\theta - \theta^*)$, where $\Omega = \dot{\theta} = |\boldsymbol{\Omega}|$. The resulting dimensionless torque is $\mathbf{m} = 2\Lambda\Omega^{-1}(\pi - \theta^*)^2\hat{\mathbf{y}}$. Balancing with a viscous drag $-\eta(\mathcal{M}\mathrm{Re})^{-1}\Omega\hat{\mathbf{y}}$ yields a steady rotation rate $\Omega = (2(\pi - \theta^*)^2\Lambda\mathcal{M}\mathrm{Re}/\eta)^{1/2}$, where $\eta \leq 15$, since part of the body sits outside of the water. A more detailed study of this rotational drag, like that performed by Hunt et al.[55], is needed.

## Discussion

Supersaturated fluids present an accessible playground for exploring the dynamics of bodies and their relationship to a complex fluid environment. In the framework proposed by Spagnolie & Underhill[56] this would appear as either a Type I or Type II system - the body is much larger than the 'obstacles' (be they bubbles or gas molecules), and the fluid exhibits a natural relaxation time. Among the unexpected findings in this system, we have observed a critical dependence of the dynamics on body rotations for large body-to-bubble size ratios, and multi-period oscillatory dynamics when a single surface interaction is insufficient to clean the body surface of its lifting agents. Another intriguing feature is that bubbles can continue to form on the body long after they cease to form at the container walls when $S_{mc} > S_{mb}$ (when the surface roughness or fibrous material scale is smaller on the container walls than on the body). Raisins inserted into a fluid which was left out for hours and appeared motionless were indeed observed to dance, albeit at a leisurely pace. Relatedly, Pereira et al.[31] identified that a smaller contact angle between a bubble and a surface decreases the energy needed to form bubbles there. Preliminary results also suggest that the body's presence can affect fluid degassing, even potentially slowing it by disturbing the large-scale convective flow responsible for gas escape, reminiscent of how moving boundaries affect flows which promote heat transport[57–60].

Additional constraints on the dynamics are expected in general. At greater depths, bubbles are less likely to grow due to the increase in hydrostatic pressure. Should a body plummet sufficiently far below the surface, bubble-assisted levitation may vanish and the plummeting will continue unresisted. Surfactants may also adjust the range of bodies possible to levitate in this manner, since their presence can affect the nature of bubble pinch-off and coalescence, in competition with the pressure and surface roughness scale via $S_{mb}$[61,62]. The shape of the container and temperature can also affect the rate of $CO_2$ loss[63]. Another uncharacterized but potentially important feature is wetting. A return to the surface releases bubbles from the body surface, but interaction with the air above may also help to nucleate other bubbles by drawing additional gas into small cavities. Initially, dry bodies danced for far longer than initially wet bodies. The interaction with the free surface appears to damp rotations as well, and as we have seen, any inhibition of the rotation of large bodies tends to inhibit vertical dancing.

A number of directions lie ahead based on additional observations not described above. Preliminary studies suggest a substantial encouragement of excursions for more elongated and asymmetric body shapes. Also, the behavior of multiple bodies in the system can result in stable rafts of bubble-sharing bodies at the surface. Since rotation is critical for triggering an excursion from the surface, and sharing bubbles inhibits rotation, the system transitions from exhibiting oscillatory to overdamped behavior with increased particle volume fraction. Additional bodies, however, also increase the fluctuations in the system, which can encourage the rolling and plummeting of others. This instability to rotation and cooperative effects are reminiscent of iceberg capsize dynamics[64–66]. Exploration of the optimal number of dancing partners is under current investigation. A more detailed study of the fluid flow itself will be highly informative regarding the nature of degassing and the effect of the immersed body.

Theoretical advances are also needed. Models of growing arrays of bubbles date back to classical works by Lifschitz and Slyozov[67] and Wagner[68]. Transient coarsening kinetics depend on numerous simultaneous mass transfer mechanisms, resulting in overlapping scaling behaviors in time[69–72]. Collective formation and dissolution of bubbles on a regular patterned grid have recently provided some insight on these coupled effects[73–75]. The additional presence of a flowing environment presents a substantial new challenge. Future work exploring body shape, multi-body dynamics, and fluid-structure interactions is likely to prove... fruitful.

## Data availability

Datasets for force growth on fixed bodies, and vertical positions during vertical oscillations, are available at https://doi.org/10.6084/m9.figshare.25302136.

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

## Acknowledgements
Support for this research was provided by the Office of the Vice Chancellor for Research and Graduate Education with funding from the Wisconsin Alumni Research Foundation, and by donations to the AMEP program (Applied Math, Engineering, and Physics), at the University of Wisconsin-Madison. S.E.S. gratefully acknowledges conversations with Hongyi Huang, Thomas G. J. Chandler, and Jean-Luc Thiffeault, and preliminary work with Carina Spagnolie.

## Author contributions
S.E.S. conceived and coordinated the research. S.E.S. and S.C. performed physical experiments, S.E.S. and C.G. performed numerical experiments. The manuscript was prepared by S.E.S.

## Competing interests
The authors declare no competing interests.
