## [Peer Review File · Nature Communications]

REVIEWER COMMENTS

Reviewer #1 (Remarks to the Author):

This is a very interesting work devoted to the motion of solid particles in a liquid pool undergoing degassing. Careful experiments are accompanied by detailed theory and sophisticated simulations. The whole presentation is built upon the experimental results and the theoretical development follows the experimental data. Three types of experiments are performed (i) to evaluate the quantity of gas leaving the liquid (ii) to measure the force on a fixed particle immersed in liquid undergoing degassing and (iii) to follow the trajectory of a free particle in liquid undergoing degassing.

The paper is well written but in some cases the density of new information is too high, so a better explanation is needed.

Definitely the work deserves publication after some clarifications:

-Please describe the difference between S_{mc} and S_{mb} .

-The procedure to evaluate the gas loss for nucleation and growth of bubbles by considering the liquid evaporation and the gas loss due to the transfer to macroscopic gas-liquid interface, is quite clever. The experiments support the quadratic law for the gas loss due to simultaneous bubble nucleation and growth. This must be further noticed.

-Equation (6) needs to be more explicit. The depended variables must be explicitly defined, and the equation must be written in a non-vectorial form.

-Please explain in more detail the differences of the results between the discrete and continuous model

- I cannot understand how the continuous model can predict rotation since it evaluates only an average coverage of the particle surface.

-It is not clear if equation (4) is a global one or it holds for each point of the particle surface.

- Figure 3 of the Supplementary material implies that bubbles appear only at the center region of the vessel. Is this so?

-According to the Supplementary Material the bubble shape is assumed hemispherical. What would be the difference in the behavior for different than $\pi/2$ contact angle?

-As I understand the drag force modification associated to the bubbles attached onto the surface of the particle is not considered. Please clarify.

-For the particular work a list of symbols is necessary.

-It is impressive how the authors present all possibilities for the extension of the present work either experimentally or theoretically.

Reviewer #2 (Remarks to the Author):

To the best of my knowledge, this article about "levitations and dynamics of bodies in supersaturated fluids" is the most in-depth in a series of more or less recent articles on this subject. For the first time a complete simulation of the phenomena is proposed which accounts for body wobbling, rolling and bouncing dynamics.

I thereby strongly suggest publication of the present manuscript in Nature Communications, provided the authors add some little precisions following the remarks below:

1) I wish the heterogeneous bubble nucleation process from supersaturated liquids could be better defined/presented in an appropriate section.

2) The walls of the glass vessel are hydrophilic, whereas the spherical body composed of PLA is hydrophobic. CO₂ repetitive bubble nucleation from nucleation sites therefore differs whether on the glass wall or on the immersed body. Could the authors detail a little bit more the phenomena at play, including the role of gas cavities acting as bubble nucleation sites.

3) I am a little bit confused with the term S_{mb} characterizing the minimum supersaturation ratio below which no bubbles can form on the immersed hydrophobic body. In my opinion, the bubble growth rate should not be equal to zero when the supersaturation ratio reaches S_{mb} , but for a value slightly lower (because of the body surface roughness)

Reviewer #3 (Remarks to the Author):

The submitted manuscript reports solid body motion in a supersaturated fluid. The solid body motion is driven by the accumulation and releasing of bubbles upon its surface. The attached bubbles can lift a mobile body upward against gravity, but arrival at a free surface can clean the body of these lifting agents and the body may plummet. Besides motile body, the authors also investigated fixed bodies to quantify fundamental features of force development and gas escape. Experimental results are

complimented with a continuum model which incorporates the dynamics of a surface buoyancy field and fluid dynamics.

In my opinion, the manuscript reports novel and interesting results which involves intricate dynamics of gas, fluid and solid. I recommend its publication and have a suggestion for the authors to consider. It is nice for the authors to carry out theoretical and numerical investigations in parallel to their experiments. However, I find that direct comparisons between experiments and theory/simulations are rare. I suggest the authors to make more connections between experiments and theory/simulations, which should help to further unveil the underlying physics.

We are very grateful to the referees for their time and effort in the review of our manuscript. Below we address all of their comments point by point.

Response to Referee 1

This is a very interesting work devoted to the motion of solid particles in a liquid pool undergoing degassing. Careful experiments are accompanied by detailed theory and sophisticated simulations. The whole presentation is built upon the experimental results and the theoretical development follows the experimental data. Three types of experiments are performed (i) to evaluate the quantity of gas leaving the liquid (ii) to measure the force on a fixed particle immersed in liquid undergoing degassing and (iii) to follow the trajectory of a free particle in liquid undergoing degassing.

The paper is well written but in some cases the density of new information is too high, so a better explanation is needed.

Definitely the work deserves publication after some clarifications:

We were very happy to see the reviewer's positive review of the paper, and belief that the work deserves publication.

-Please describe the difference between S_{mc} and S_{mb} .

We have added comments to better highlight the meaning of and difference between S_{mc} and S_{mb} , the minimal values of the supersaturation ratio below which bubbles cease to form on the container, and on the body, respectively. Since bubble growth depends on the nature of the surface, either due to its roughness, or to other particulate matter there which is common, there is generally a different cutoff gas concentration for bubble growth on the container, and on the body surface. In this work, bubbles cease to grow on the container long before they stop growing on the body, though this could be reversed with different materials.

-The procedure to evaluate the gas loss for nucleation and growth of bubbles by considering the liquid evaporation and the gas loss due to the transfer to macroscopic gas-liquid interface, is quite clever. The experiments support the quadratic law for the gas loss due to simultaneous bubble nucleation and growth. This must be further noticed.

We now point out the encouraging success of the quadratic scaling.

-Equation (6) needs to be more explicit. The depended variables must be explicitly defined, and the equation must be written in a non-vectorial form.

We have tried to rearrange this section, with the hope that it is a smoother entry for the reader. On the reviewer's second point, we believe that the equations in vector form are clearer than they would be in a different form, but we acknowledge that this may be a matter of taste. We have chosen to leave the equations in vector form.

-Please explain in more detail the differences of the results between the discrete and continuous model

We have added additional commentary on the relationship between the discrete and continuum model results.

- I cannot understand how the continuous model can predict rotation since it evaluates only an average coverage of the particle surface.

The continuous model assigns a lifting traction on each point of the body surface based on a *local* average of bubbles there. There can still be regions of the surface that have many bubbles, and other parts of the surface which have no bubbles. This configuration generically produces a torque on the body. We have added a few clarifying notes to this effect, and we now make the spatial dependence explicit in Eqs. (4) and (8) as a helpful reminder.

-It is not clear if equation (4) is a global one or it holds for each point of the particle surface.

We should have stated the equation more clearly. Hopefully the adjustments above now clear up any potential for uncertainty.

- Figure 3 of the Supplementary material implies that bubbles appear only at the center region of the vessel. Is this so?

This is not generally true, the simple model is proposed because it is analytically tractable, and is expected to give qualitatively similar results to more complicated situations (we are currently working on that question but it is a separate adventure). We now stress that this is a very coarse approximation. That said, for such a rough estimate, that we estimate the real experimental result to within a factor of 2/3 (and not, say, within one or two orders of magnitude) was quite satisfying to us.

-According to the Supplementary Material the bubble shape is assumed hemispherical. What would be the difference in the behavior for different than $\pi/2$ contact angle?

This question has recently been considered by Pereira et al. (R. Soc Open Sci. 2023). They find that as the (bubble-surface) contact angle decreases, so too does the energy required for a bubble to nucleate. This would then prolong the bubble growth on the body, and the time in which to observe the dancing dynamics. We have added a couple notes in the manuscript on this point.

-As I understand the drag force modification associated to the bubbles attached onto the surface of the particle is not considered. Please clarify.

It is true that we did not model the different boundary conditions for a bubble-laden body. We now acknowledge this as a potential direction for improvement in the paper when we introduce the model. Since the transit time between the free surface and container floor is small compared to the residence times there, we are quite confident (based on other largely uninteresting numerical simulations) that this is a safely neglected aspect of the physics. We do now acknowledge that we are neglecting this detail in the manuscript.

-For the particular work a list of symbols is necessary.

After discussing this with the journal editor to determine the appropriateness of a glossary in the paper, we have settled on a compromise which we hope the reviewer and journal will find acceptable. A glossary of symbols now appears in the Supplemental Information.

-It is impressive how the authors present all possibilities for the extension of the present work either experimentally or theoretically.

We believe that the system invites much more consideration. There is much to be done!

Sincerely,

Saverio E. Spagnolie

Sam Christiansen

Carsen Grote

We are very grateful to the referees for their time and effort in the review of our manuscript. Below we address all of their comments point by point.

Response to Referee 2

To the best of my knowledge, this article about "levitations and dynamics of bodies in supersaturated fluids" is the most in-depth in a series of more or less recent articles on this subject. For the first time a complete simulation of the phenomena is proposed which accounts for body wobbling, rolling and bouncing dynamics.

I thereby strongly suggest publication of the present manuscript in Nature Communications, provided the authors add some little precisions following the remarks below:

We were very happy to see the reviewer's strong suggestion for publication.

1) I wish the heterogeneous bubble nucleation process from supersaturated liquids could be better defined/presented in an appropriate section.

This is a big question, and we do not have (or are not aware) of an adequate model for the complete process that is sufficiently well matched to the experiment. Transient Ostwald ripening, which includes in the mass transfer process not only diffusion-based growth, but also coalescence, and detachment, and a background flow, and surface wetting, is not something that we have found in the literature. Note that each aspect contributes different scaling behaviors, so transients are likely extremely complicated to analyze cleanly. We have made this a bit clearer in the discussion section, where we nudge (others? our future selves?) that further theoretical development is needed.

2) The walls of the glass vessel are hydrophilic, whereas the spherical body composed of PLA is hydrophobic. CO₂ repetitive bubble nucleation from nucleation sites therefore differs whether on the glass wall or on the immersed body. Could the authors detail a little bit more the phenomena at play, including the role of gas cavities acting as bubble nucleation sites.

The nucleation from the glass wall is believed to be dominated not by interactions with a clean surface, where hydrophobicity/hydrophilicity would be relevant, but instead in small fibrous material left behind during cleaning. The 'lumen' of these fibrous materials are remarkable sites for consistent and rapid bubble growth and release, since a pocket of gas remains behind after each pinch-off event, ready to charge the next bubble. It is also why bubble production appears to be confined to a few points on the surface, rather than being more uniformly distributed. Nevertheless, there is still a supersaturation ratio below which bubbles will cease to form in these lumen - in the paper this value is called S_{mc} . But the reviewer spurs us to reword its introduction so that it more broadly captures the value below which bubbles are not produced on the container *by any means*.

There is indeed a substantially different process occurring on the body itself, which we model as having a uniform production, and which is accounted for using a different constant, S_{mb} for the minimal supersaturation ratio needed for production.

3) I am a little bit confused with the term S_{mb} characterizing the minimum supersturation ratio below which no bubbles can form on the immersed hydrophobic body. In my opinion, the bubble growth rate should not be equal to zero when the supersaturation ratio reaches S_{mb} , but for a value slightly lower (because of the body surface roughness)

This is precisely the definition of S_{mb} , the value below which no bubbles form - which *includes* all details such as surface roughness. We have added a comment to the paper to make sure this is clearer.

Sincerely,

Saverio E. Spagnolie
Sam Christiansen
Carsen Grote

We are very grateful to the referees for their time and effort in the review of our manuscript. Below we address all of their comments point by point.

Response to Referee 3

The submitted manuscript reports solid body motion in a supersaturated fluid. The solid body motion is driven by the accumulation and releasing of bubbles upon its surface. The attached bubbles can lift a mobile body upward against gravity, but arrival at a free surface can clean the body of these lifting agents and the body may plummet. Besides motile body, the authors also investigated fixed bodies to quantify fundamental features of force development and gas escape. Experimental results are complimented with a continuum model which incorporates the dynamics of a surface buoyancy field and fluid dynamics.

In my opinion, the manuscript reports novel and interesting results which involves intricate dynamics of gas, fluid and solid. I recommend its publication and have a suggestion for the authors to consider. It is nice for the authors to carry out theoretical and numerical investigations in parallel to their experiments. However, I find that direct comparisons between experiments and theory/simulations are rare. I suggest the authors to make more connections between experiments and theory/simulations, which should help to further unveil the underlying physics.

We were pleased to see the reviewer's recommendation for publication.

We agree that comparison of experiments and theory is rare. We are also a bit confused by the suggestion, in that we believe the paper represents a rather substantial comparison already! The solid/dashed curves in Figs. 3c-d, and Fig. 4e are all theoretical estimates, which pass very closely to the experimental results using a much smaller number of fitting parameters than there were experimental measurements. The closeness of the match in measuring the force on the stationary body is what gave us confidence to perform numerical experiments for the vertical oscillation (dancing) frequency, and then those naturally fell right on top of separate experimental results, as shown in Fig 4e. The one area where we are not yet attempting the comparison is in the rotational dynamics. This is certainly of interest to us, but we are not able to do these experimental measurements in the lab (yet - this is a work in progress and is perhaps a year away). It is certainly on our long list of questions for the future. In the revised manuscript we have added a few comments in an effort to better identify these connections, and to indicate this future goal.

Sincerely,

Saverio E. Spagnolie
Sam Christiansen
Carsen Grote

REVIEWERS' COMMENTS

Reviewer #1 (Remarks to the Author):

I am satisfied with the replies of the authors and the revised manuscript. I recommend publication of the revised manuscript.

Reviewer #2 (Remarks to the Author):

This revised version is clarified and the points raised were carefully taken into account.

I therefore recommend publication of the present manuscript

Reviewer #3 (Remarks to the Author):

The authors have successfully addressed my question. The manuscript can be published in its current form.